# NSM4D: Neural Scene Model Based Online 4D Point Cloud Sequence Understanding

## Abstract

Understanding 4D point cloud sequences online is of significant practical value in various scenarios such as VR/AR, robotics, and autonomous driving. The key goal is to continuously analyze the geometry and dynamics of a 3D scene as unstructured and redundant point cloud sequences arrive. And the main challenge is to effectively model the long-term history while keeping computational costs manageable. To tackle these challenges, we introduce a generic online 4D perception paradigm called NSM4D. NSM4D serves as a plug-and-play strategy that can be adapted to existing 4D backbones, significantly enhancing their online perception capabilities for both indoor and outdoor scenarios. To efficiently capture the redundant 4D history, we propose a neural scene model that factorizes geometry and motion information by constructing geometry tokens separately storing geometry and motion features. Exploiting the history becomes as straightforward as querying the neural scene model. As the sequence progresses, the neural scene model dynamically deforms to align with new observations, effectively providing the historical context and updating itself with the new observations. By employing token representation, NSM4D also exhibits robustness to low-level sensor noise and maintains a compact size through a geometric sampling scheme. We integrate NSM4D with state-of-the-art 4D perception backbones, demonstrating significant improvements on various online perception benchmarks in indoor and outdoor settings. Notably, we achieve a **9.6%** accuracy improvement for HOI4D online action segmentation and a **3.4%** mIoU improvement for SemanticKITTI online semantic segmentation. Furthermore, we show that NSM4D inherently offers excellent scalability to longer sequences beyond the training set, which is crucial for real-world applications.

## 1 Introduction

The utilization of 4D point cloud sequences, which combine 3D spatial information with temporal dynamics, has become integral to numerous modern AI applications, such as robotics, autonomous driving, and AR/VR. These sequences offer a distinct advantage by faithfully capturing the geometry and motion of dynamic scenes. Consequently, the online understanding of such 4D data has emerged as a crucial task, necessitating the persistent and consistent comprehension of scene geometry and dynamics based on only current and historical observations.

While advancements have been made in autonomous driving through the use of domain-specific representations like 2D BEV maps or domain-specific knowledge like known ego-motion, a significant gap remains for a generic framework capable of addressing scenarios that defy these priors. For generic online 4D perception, existing works usually treat point cloud sequences as unstructured 4D data (Liu et al., 2019; Fan & Yang, 2019; Fan et al., 2021; Wen et al., 2022; Garcia & de Queiroz, 2017; Fan et al., 2022; He et al., 2022; Shi et al., 2022) without exploiting the prior that such points are sampled from 3D geometry moving following some low-dimensional trajectories. Current point cloud observation directly queries features from raw point clouds at different time stamps in history whose motion and geometry are coupled. This can be hard to optimize and usually leads to less effective 4D features. Moreover, existing works usually leverage a fixed-length window to restrict the temporal query scope for computation feasibility. This also prevents online 4D perception from accessing long-term spatial-temporal context.

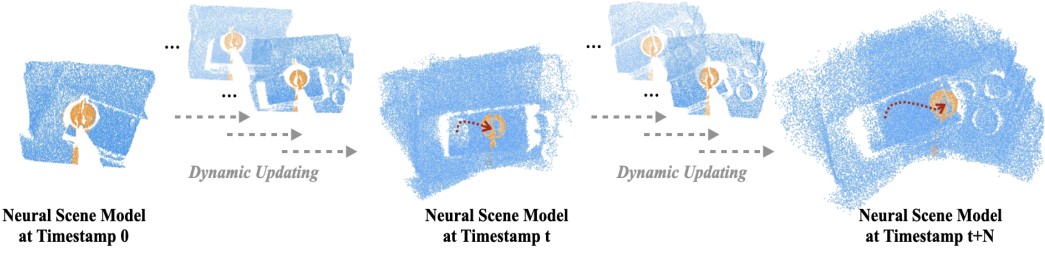

Figure 1: NSM4D employs geometry and motion factorization to carry out implicit reconstruction. The neural scene model is recurrently constructed through dynamic updating, effectively aggregating historical observations to form a comprehensive scene while keeping track of the motion information.

The current challenges of generic online 4D perception mainly come from two aspects. First, the 4D point cloud sequence usually couples 3D geometry and its dynamic motion, forming a redundant and noisy data form in a high dimensional space. This could lead to a severe learning issue. Second, it is crucial to track the geometry and motion information over an extended history as they could be useful at any time. However, retrieving this historical data may suffer from computation explosion as the sequence grows longer and longer without careful design.

To address the aforementioned challenges, we introduce a novel paradigm named NSM4D, which serves as a plug-and-play technique adaptable to existing 4D backbones, enabling them to achieve effective and efficient online 4D perception capabilities. Our approach is grounded in the insight that the information within a 4D sequence can be efficiently factorized into geometry and motion through dynamic 3D reconstruction, significantly reducing data redundancy while facilitating the incorporation of long-term history. Querying a dynamic reconstruction for historical spatial-temporal contexts would be much more efficient compared with directly querying the raw 4D data. However, existing dynamic reconstruction techniques lack reliability and efficiency when dealing with noisy point cloud sequences. To overcome this limitation, we propose to reconstruct the scene implicitly, forming a neural scene model that consists of a set of geometry tokens equipped with factorized geometry and motion features. The neural scene model is constructed recurrently through a dynamic updating method as shown in Figure 1. At each time step, upon the arrival of a new point cloud observation, we leverage a token updating module to align the neural scene model with the current observation. In particular, we estimate a scene flow from the previous time step and employ it to shift the tokens and update the motion features of the existing neural scene model. Concurrently, we exploit a token generation module to extract token-wise geometry and motion features from the current observation and further integrate the new tokens into the neural scene model through a token merging module. The token merging module efficiently updates the neural scene model while maintaining its moderate size. As a result, the neural scene model keeps track of the point cloud sequence and provides a compact and dynamic summary of the up-to-date history for online perception. Furthermore, the token-level representation enhances reconstruction robustness against sensing noise and scene flow estimation errors, resulting in a lightweight, flexible, and expressive dynamic memory.

We extensively evaluate our approach on several benchmarks including both indoor and outdoor scenarios. We find that NSM4D significantly boosts the online perception ability of the state-of-the-art 4D perception models(+9.6% accuracy on HOI4D action segmentation, +1.9% mIoU on HOI4D semantic segmentation, +3.4% mIoU on SemanticKITTI). We also observe that NSM4D demonstrates exceptional scalability, extending its effectiveness to longer sequences that surpass the length of the training sequences.

Our contributions are threefold: First, we propose a new paradigm that serves as a plug-and-play strategy to existing 4D backbones by compressing the historical point cloud sequence into a compact neural scene model to boost their ability of online 4D perception. Second, we design a token-based neural scene model by factorizing the geometry and motion information from a point cloud sequence and present a dynamic updating method to recurrently construct the model. Third, we conduct extensive experiments and analysis, the strong results show the effectiveness of NSM4D and its generalizability to different tasks and both indoor and outdoor scenarios.

## 2 RELATED WORK

**Point Cloud Sequence Processing** holds great importance for enabling intelligent agents to comprehend the dynamic nature of our 3D world. Existing methods can be divided into two categories: voxel-based (Lin et al., 2018; Wang et al., 2020; Choy et al., 2019) and point-based (Fan & Yang, 2019; Fan et al., 2021; 2022; Wen et al., 2022; Zhong et al.; Rempe et al., 2020; Wei et al., 2022; Min et al., 2020). For voxel-based methods, MinkowskiNet (Choy et al., 2019) apply 4D convolution to extract spatio-temporal information from voxel grids. SpSequenceNet (Shi et al., 2020) utilizes KNN to aggregate 4D spatio-temporal information on both global and local level. SVQNet (Chen et al., 2023) constructs voxel-adjacent network to take full advantage of history knowledge. For the point-based method, PSTNet (Fan et al., 2022) proposes point spatio-temporal (PST) convolution to reach informative representations of point cloud sequences. P4Transformer (Fan et al., 2021) employs a transformer to avoid point tracking for raw point cloud sequence modeling. PPTr (Wen et al., 2022) introduces primitive fitting and memory pool to integrate long-term information. These methods primarily concentrate on offline settings, exhibiting limited capability when it comes to adapting to online scenarios. Therefore, we propose NSM4D, which is a plug-and-play strategy that can be adapted to existing 4D backbones, significantly improving their online perception capability.

**Online Perception** holds the key to real-world applications. In real application scenarios, the entire sequence is not accessible and predictions are required solely based on historical observations. Many methods focusing on online perception have been proposed for practical application needs. VISOLO (Han et al., 2022) utilizes grid-structured representation for effective instance segmentation. Colar (Yang et al., 2022) introduces exemplar-consultation mechanism to formulate an efficient network for action segmentation. DVIS (Zhang et al., 2023) incorporates referring tracker and temporal refiner to decouple online instance segmentation into subtasks. (Mersch et al., 2022) exploits 2D range image representation to perform online point cloud prediction. Existing works for online perception mainly focus on 2D video with less work focusing on 4D point cloud sequences. Online understanding of 4D point cloud sequences poses challenges due to the unstructured and redundant data format. Therefore, we propose NSM4D, a novel plug-and-play module that offers a generic paradigm for online processing of 4D point cloud sequences.

**Memory Mechanism.** Memory mechanism has been substantially studied in different tasks. Such mechanisms are mostly designed to tackle long sequences and dense input. Considering the variations of data, the proposed memory mechanisms are mainly used in two scenarios: indoor scene and outdoor scene. For indoor scenarios, SMT (Fang et al., 2019) incorporates global scene memory to facilitate policy generation. PointRNN (Fan & Yang, 2019) introduces a 4D recurrent neural network to effectively aggregate historical point features and caches them. PPTr (Wen et al., 2022) computes primitive-level representations of long-range videos and maintains an offline memory pool. For outdoor scenarios, BEVFormer (Li et al.) introduces BEV maps and multi-stage attention to exploit spatio-temporal information from historical multi-camera images. These methods either use a global feature for historical frames, or directly integrate some basic memory models for historical information aggregation. Others utilize strong domain prior knowledge in a more convenient way but at the cost of losing versatility. Our proposed NSM4D not only aggregates historical geometry and motion information in a more synergetic and compact way but also maintains the generalization ability on both indoor and outdoor scenarios.

## 3 PROBLEM STATEMENT

In the context of a point cloud sequence, online 4D perception refers to the task of understanding the state or forecasting the behavior of the point cloud as each new frame arrives. In this paper, our primary focus lies on two tasks: online action segmentation and online semantic segmentation. In the context of action segmentation, the objective is to assign an action label to each processed frame. While in semantic segmentation, the goal is to assign a semantic label to each individual point.

When comparing online perception to offline perception, there are two main differences that consequently introduce specific challenges. First, in online perception, **predictions have to be provided frame by frame** instead of all at once. We need to integrate the information from early timestamps to enhance contextual understanding while maintaining computational efficiency. So effectively and efficiently caching computations from early timestamps is of great importance. However, offline

perception is not bound by this constraint. Second, online perception needs to deal with **temporal observations of varying lengths** from the history. Short history presents challenges in accurately modeling the complete geometry of the scene, whereas a long history poses greater difficulties in modeling the motion of the scene. However, offline perception has a more homogeneous temporal context and should be more friendly to learn and optimize. Our design focuses on adapting existing offline backbones to mitigate the gap.

## 4    NEURAL SCENE MODEL

Our goal is to devise a methodology that is both effective and efficient for sequentially aggregating long-term historical information for precise 4D online perception. NSM4D aims to achieve this by introducing a generic paradigm design that enables seamless integration with the existing 4D networks, empowering them with the capability of efficient and effective online 4D perception.

There are three key challenges in our design. The first challenge arises from the redundant and noisy nature of the point cloud sequence, necessitating the compression into a more compact and robust representation. An effective way is to decouple geometry and motion to reduce the representation into a low-dimensional space. This separation is advantageous as it allows for distinct depiction of the dynamic scene by capturing the specific aspects of geometry and motion. Many previous approaches have mixed these two aspects, potentially resulting in the loss or confusion of their distinct characteristics. The second challenge pertains to the dynamic nature of the coordinate system, which constantly changes due to the unknown varying camera poses. Aligning the features of different frames within this changing coordinate system presents a significant challenge, rendering the compression of the point cloud sequence difficult. The third challenge involves ensuring a lightweight compression of historical information, which is memory-friendly while also capable of preserving long-term historical information. We propose NSM4D, which constructs a neural scene model to preserve geometry tokens separately storing geometry and motion features to compress past scene information and leverage estimated scene flows to dynamically deform and update the neural scene model when new frames are encountered.

Existing generic 4D backbones generally include two modules: **local feature extraction** to process point clouds into a set of features, **spatial-temporal context aggregation** that usually encodes coupled geometry and motion information to provide a context of the frame of interest. To achieve seamless adaptation, we rely on the first module as our grounding point, while substituting the second module with NSM4D.

### 4.1    OVERVIEW

An overview of our dynamic updating method is shown in Figure 2, which consists of three modules. The **4D Tokenization** module extracts token-wise geometry and motion features from the current observation. It utilizes the local feature extractor in the 4D backbone we rely on to extract point/voxel features. Simultaneously, an auxiliary flow estimator is introduced to encode point motion features independently. Next, geometry tokens are sampled throughout the frame, and these tokens are utilized to perform set abstraction, resulting in token-wise geometry and motion features. This process enables NSM4D to acquire more compact and lightweight features, greatly facilitating subsequent scene model updates. The **Token Updating** module is employed to deform the scene model in order to align the neural scene model with the current observation. This is achieved by utilizing the estimated scene flow to shift the geometry tokens and update the motion features of the existing neural scene model, resulting in a deformed model. Next, we integrate the new tokens from the current frame into the deformed neural scene model using the **Token Merging** module. To preserve the model size, this module also incorporates a sample convolution to control the token numbers. And we will get a deformed and updated neural scene model at timestamp t now. Finally, a cross-attention module is applied between the current frame's feature and the neural scene model feature to efficiently retrieve useful history information to generate accurate dense predictions.

### 4.2    4D TOKENIZATION

We choose to decouple geometry and motion exploiting the prior that points in the sequence are sampled from 3D geometry and move along low-dimensional trajectories. This approach should be

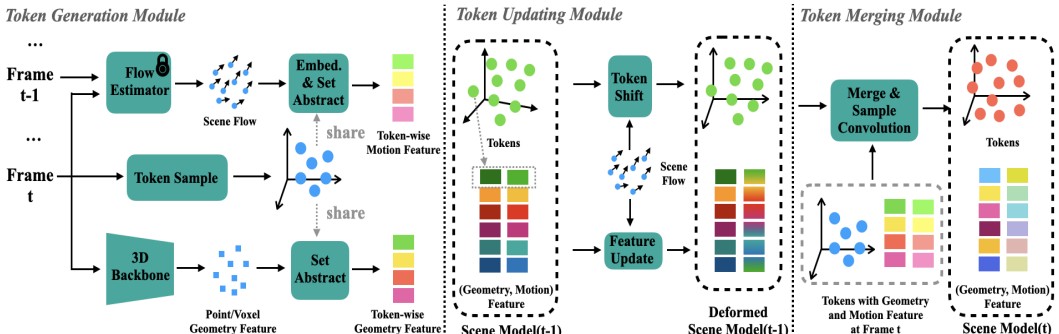

Figure 2: Overview of NSM4D. (a) **4D Tokenization** module samples tokens and extracts token-wise geometry and motion features from the current observation. (b) **Token Updating** module utilizes scene flow to deform the scene model by shifting the tokens and updating the motion feature, ensuring alignment with the current observation. (c) **Token Merging** module integrates new tokens to efficiently update the neural scene model while maintaining its moderate size.

an effective way to compress the point cloud sequence to reduce redundancy and noise. Additionally, geometry and motion contain distinct information that contributes to the depiction of the 4D scene. Therefore, it is crucial to consider these aspects separately.

For the geometry feature, we use the set abstraction technique to sample anchor points to extract token-wise geometry features, regardless of whether the network NSM4D grounds on is built upon point features, voxel features, or any other feature representation. For motion features, point-wise scene flow is a good indicator of the motion information of the dynamic scene. To obtain the point-wise motion feature, we encode the scene flow calculated by the state-of-the-art flow estimator RAFT3D(Teed & Deng, 2021). Subsequently, we employ the same anchor points to perform set abstraction, resulting in the generation of token-wise motion features.

**Token-wise Geometry Feature.** For each frame, we sample $r_s$ anchor points $T^R = \{\alpha\}_i^{r_s}$ among a set of features extracted by the base network. For each anchor point $\alpha$, we group its neighbor feature and utilize set abstraction as introduced in PointNet++(Qi et al., 2017) to construct the token-wise geometry feature $F^\alpha$. Uniformly sampled tokens with geometry features $G^R = \{(\alpha, F^\alpha)\}_{i=1}^{r_s}$ represent the scene geometry of the frame in a compact way.

Abstracting the geometry feature to the token-level offers two key advantages. First, it results in a more compact and robust representation compared to point-level or voxel-level representations. Each token represents the geometry of a local region, leading to enhanced efficiency and robustness in constructing the scene model. Second, as we will discuss further, shifting the geometry tokens to a new coordinate system requires the use of scene flow. Averaging the token-wise scene flow should be more robust than using point-wise or voxel-wise representations, as the latter are more susceptible to noise interference.

**Token-wise Motion Feature.** Movement trajectory faithfully depicts the historical motion information. However, tracking the trajectory of each point in the 4D scene is highly difficult, with two main challenges. First, the coordinate system constantly changes between frames, so the observed motion is a mixture of actual object motion and camera ego-motion, which is not easy to factorize. Second, there are points in and out between frames, making it unfeasible to track all of them.

To alleviate the above challenges, making some approximations is reasonable. Thus, we use scene flow as an alternative to actual motion. Although the scene flow mixes the object motion and camera ego-motion, it still contains information on rough moving direction and speed, which are also crucial cues. Accumulation of scene flows is an approximation of movement trajectory. In addition, we utilize the scene flow to shift geometry tokens, as will be discussed in detail in the next section.

We use RAFT3D(Teed & Deng, 2021) to estimate the scene flow $F^R = \left\{\tilde{f}^i\right\}_{i=1}^N$. Specifically, RAFT3D estimates the dense flow field by iteratively updating the SE3 field with the constructed 4D correlation volume. We can get a relatively accurate estimation at large throughput rates with

a good balance of performance and speed. Subsequently, a shared MLP embedding layer followed by a recurrent network is employed to construct the point motion feature. Finally, the same anchor points $T^R = \{\alpha\}_{i=1}^{r_s}$ are utilized to perform set abstraction to extract token-wise motion features $M^R = \{(\alpha, D^\alpha)\}_{i=1}^{r_s}$.

### 4.3 TOKEN UPDATING

To accommodate the incoming frame, we shift the scene model to the new coordinate system to ensure alignment. Achieving alignment between the frame of interest and the memory is a non-trivial task. However, leveraging the estimated scene flows and geometry tokens, we make this alignment feasible and attainable. Below, we present a formal illustration of the process.

**Token Shift.** $T^W = \{\omega\}_{i=1}^n$ represents the geometry tokens of the scene model where $\omega$ is the token coordinate. Upon the arrival of a new point cloud frame, we shift the geometry tokens from the last frame coordinate system to the current frame one and get $\hat{T}^W = \{(\omega + f^\omega)\}_{i=1}^n$, where $f^\omega$ is the token-level scene flow averaged from the point-level scene flow $F^R$.

During the token shift process, we simply copy-and-paste the token-wise geometry feature to encourage the network to learn the invariant geometry feature. Shifted tokens with geometry feature is $\hat{G}^W = \{(\omega + f^\omega, F^\omega)\}_{i=1}^n$ now. We do not consider the equivariance property of the feature here due to it will introduce further computation overhead and the robustness of the equivariance is also not guaranteed.

**Token-wise Motion Feature Update.** After shifting the tokens, the motion feature still stays on characterizing the motion information before the current frames. We need to update the motion feature to incorporate new motion as the tokens have been shifted. Specifically, we leverage the shifted tokens to perform set abstraction on the point motion feature of the current frame. We then concatenate the extracted token features with the existing ones. Finally, an MLP is employed to fuse the features and carry out the necessary updates. We represent the updated motion features with their corresponding shifted tokens as $\hat{M}^W = \{(\omega + f^\omega, D^\omega)\}_{i=1}^n$.

### 4.4 TOKEN MERGING

Having shifted the neural scene model to the current frame, we have successfully aligned the tokens and features with the current frame. Consequently, merging the new tokens and features into the scene model to incorporate the information from the new frame becomes a straightforward process. After merging, the tokens $T^W$ is expanded to $T^W + T^R$, the geometry feature $\hat{G}^W$ is expanded to $\hat{G}^W + G^R$, and the motion feature $\hat{M}^W$ is expanded to $\hat{M}^W + M^R$. Next, we execute sample convolution modules $S^G$ and $S^M$ on the merged geometry features and motion features, respectively, employing set abstraction techniques to regulate the number of tokens. We will get new neural scene model tokens with new token-wise geometry features $G^W = S^G(\hat{G}^W + G^R)$, and motion features $M^W = S^M(\hat{M}^W + M^R)$. This maintains the size of the scene model and yields the new tokens with geometry and motion features.

### 4.5 NEURAL SCENE MODEL QUERY

With the neural scene model, which has already incorporated all required historical information, we can directly interact with it to get valuable historical cues for the current dense prediction. We achieve this through the cross-attention mechanism. It is done by setting the current frame's geometry token feature as the query and the scene model's geometry and motion token feature as the key. The whole attention process is illustrated in the following equation.

$$Q = W_Q \cdot G^R, K_G = W_{K_G} \cdot G^W, V_G = W_{V_G} \cdot G^W$$
$$K_M = W_{K_M} \cdot M^W, V_M = W_{V_M} \cdot M^W$$
$$\text{Feature} = \text{Softmax}\left(\frac{QK_G^T}{\sqrt{d_k}}\right) V_G + \text{Softmax}\left(\frac{QK_M^T}{\sqrt{d_k}}\right) V_M$$

Subsequently, distinct prediction heads are employed for action and semantic segmentation, leading to the generation of final prediction results. In the case of action segmentation, we utilize a maximum operation on the tokens to extract a global feature, which is then processed by an MLP to obtain the final classification scores for the current frame. For semantic segmentation, we incorporate a decoder with deconvolution that performs upsampling to obtain per-point classification scores.

## 5 EXPERIMENTS

In this section, we apply NSM4D as a plug-and-play strategy to modern 4D networks to equip them with a strong ability for online 4D perception. We show experiment results on several indoor and outdoor benchmarks.

### 5.1 DATASETS AND METRICS

**HOI4D** (Liu et al., 2022) is a large-scale 4D egocentric dataset with rich annotations, to catalyze the research of category-level human-object interaction. It consists of 3863 point cloud sequences by 9 participants interacting with 800 different object instances from 16 categories over 610 different indoor rooms, with 150 frames in each sequence. We follow the official data split of HOI4D with 2971 training scenes and 892 test scenes. In terms of HOI4D action segmentation, it offers action labels for each frame, encompassing a total of 19 distinct classes. For HOI4D semantic segmentation, it provides per-point labels with a total of 39 classes.

**SemanticKITTI** (Behley et al., 2019) is a large-scale outdoor dataset for autonomous driving applications. The data was captured using a 64-beam LiDAR sensor. There are totally 22 sequences. The semantic segmentation task in this dataset is officially divided into two phases. The first phase, known as the single-scan phase, involves labeling 19 semantic classes without differentiating between moving and static objects. The second phase, referred to as the multi-scan phase, expands the semantic segmentation task to include 25 semantic classes distinguishing between moving and static objects. We select the multi-scan task as the validation benchmark for NSM4D, as this task requires the model to effectively aggregate information from multiple frames in order to differentiate between moving and static objects. This choice of task allows for a clear demonstration of NSM4D's online perception capability.

**Metrics.** For action segmentation, we report the following metrics: framewise accuracy (Acc), segmental edit distance, as well as segmental F1 scores at the overlapping thresholds of 10%, 25%, and 50%. Overlapping thresholds are determined by the IoU ratio. For semantic segmentation, the mean Intersection of Union(IoU) is used as the evaluation metric.

**Implementation Details.** As NSM4D is a plug-in strategy, we ground NSM4D on three existing 4D networks including P4Transformer (Fan et al., 2021) and PPTr (Wen et al., 2022) for indoor scenarios, and Point Transformer V2 (Wu et al., 2022) for outdoor scenarios. In HOI4D action segmentation, we adopt a sequence length of 150, consistent with prior work, but with some minor modifications that will be elaborated on later. Additionally, we employ longer sequence lengths to assess the scalability of NSM4D. As for HOI4D semantic segmentation and SemanticKITTI, we utilize a sequence length of 3, following a conditional setting. All models are trained in an end-to-end fashion using a frozen flow estimator on an NVIDIA A100 GPU. For action segmentation and semantic segmentation, frame-wise and point-wise cross-entropy losses are employed individually.

### 5.2 INDOOR DATASETS

**HOI4D online action segmentation**. HOI4D dataset provides sequences with a fixed length of 150. However, longer sequences are more common for real-world scenarios. We also observe that training on the former fixed-pattern data (similar start and end action) leads to significant overfitting and has weak generality to sequences with a random starting scene, reflecting the irrationality of data organization. To alleviate these problems, we prepare a new dataset leveraging data from the HOI4D action segmentation dataset. Specifically, given sequences with 150 frames in HOI4D, we flip the sequence for a reverse sequence of 150. Then we stitch together the original and reverse sequences to get longer sequences. We follow the setup of HOI4D and preserve a training set with 2971 sequences

| Method | Frames | Acc | Edit | F1@10 | F1@25 | F1@50 |
|---|---|---|---|---|---|---|
| P4Transformer | 150 | 52.65 | 39.71 | 42.42 | 36.89 | 26.49 |
| + **NSM4D**(Ours) | 150 | 58.74 | 47.88 | 51.67 | 44.14 | 35.79 |
| PPTr | 150 | 61.73 | 49.19 | 53.08 | 48.49 | 38.56 |
| + **NSM4D**(Ours) | 150 | 67.78 | 63.17 | 68.01 | 63.71 | 51.85 |
| + **NSM4D**(Ours) | 600 | **71.31** | **67.95** | **72.14** | **68.09** | **56.46** |

| Method | Frames | mIoU |
|---|---|---|
| P4Transformer | 3 | 41.61 |
| + **NSM4D**(Ours) | 3 | 42.54 |
| PPTr | 3 | 42.73 |
| + **NSM4D**(Ours) | 3 | 44.04 |
| + **NSM4D**(Ours) | 10 | **44.67** |

Table 1: **Left**. Online action segmentation results on HOI4D. **Right**. Online semantic segmentation results on HOI4D.

| Method | mIoU | car | bic. | mot. | tru. | ove. | per. | roa. | par. | sid. | bui. | fen. | veg. | trm. | ter. | pol. | tra. | mca. | mbi. | mpe. |
|---|---|---|---|---|---|---|---|---|---|---|---|---|---|---|---|---|---|---|---|---|
| MinkUnet | 48.47 | 93.8 | 23.7 | 48.9 | **90.3** | 41.3 | 18.0 | 92.2 | 32.2 | 78.4 | 89.8 | 55.5 | 88.8 | 63.7 | 77.0 | 63.6 | 50.0 | 69.2 | 83.1 | 52.5 |
| SparseConv | 48.99 | 94.7 | 24.1 | 54.1 | 69.6 | 43.4 | 17.3 | 93.2 | 45.1 | 79.8 | 89.5 | **61.7** | 87.7 | 62.9 | 74.6 | 63.8 | 50.0 | 73.9 | 85.4 | 53.6 |
| SPVCNN | 49.70 | 93.9 | 34.4 | 64.7 | 68.0 | 33.0 | 19.7 | 93.6 | 45.2 | 80.1 | 90.3 | 59.7 | 88.4 | 63.5 | 75.6 | 64.1 | 51.9 | 74.3 | **86.7** | 55.0 |
| PTv2 | 51.13 | 91.3 | 52.8 | 70.1 | 79.3 | 58.3 | 19.2 | 93.7 | 55.0 | 78.6 | 90.0 | 55.0 | 89.8 | 66.2 | 78.1 | 63.5 | 54.3 | 40.7 | 84.0 | 55.5 |
| + **NSM4D**(Ours) | **54.48** | **96.3** | **56.4** | **71.5** | 83.1 | **60.4** | **24.6** | **94.7** | **59.1** | **81.8** | **91.5** | 57.7 | **90.2** | **67.9** | **79.1** | **65.6** | **54.7** | **79.4** | 86.2 | **61.5** |

Table 2: Multi-scan semantic segmentation results on SemanticKITTI.

and a test set with 892 sequences. Each sequence starts with a randomly sampled index. There are 22 classes now due to new actions appearing in the flipped sequence.

As reported in the left of Table 1, it introduces a large margin of improvement over all the reported metrics when introducing NSM4D into the vanilla models on the length of 150, demonstrating the strong ability of the neural scene model to aggregate the historical geometry and motion information. As our scene model should benefit from long-term information, we apply it to longer sequences. As shown in the table, the performance increases as the sequence length gets longer, showing the capability of NSM4D to maintain long-term historical information. However, the vanilla models do not exhibit such properties, which we will have a more detailed discussion regarding the long sequence scalability in Section 5.4.

**HOI4D online semantic segmentation**. As reported in the right of Table 1, it shows significant improvement when introducing NSM4D into the vanilla models. It further demonstrates the strong ability of NSM4D to maintain and leverage spatio-temporal information for dense prediction task. Further performance is observed when enlarging the sequence length, demonstrating the long sequence scalability of NSM4D.

### 5.3 OUTDOOR DATESTES

**SemanticKITTI online semantic segmentation.** As shown in Table 2, with the incorporation of NSM4D, vanilla models gain a significant improvement. This strongly indicates the generalizability of NSM4D to dense prediction tasks in outdoor scenarios. Furthermore, the notable improvement on moving objects and small objects, when compared to vanilla models, serves as a compelling example of NSM4D's capability to effectively aggregate historical geometry and motion information to serve the current frame.

### 5.4 ANALYSIS

To provide more insights of NSM4D, we conduct several analysis experiments on HOI4D action segmentation. We primarily focus on discussing the scalability of our method and demonstrating the effectiveness of geometry and motion factorization. More additional ablation studies can be found in the appendix.

**Long sequence scalability.** In real-world scenarios, the length of point cloud sequences can vary greatly, depending on the application and the environment. A robust model should possess the ability to adapt to various sequence lengths, particularly when dealing with very long sequences. Nonetheless, due to memory limitations, training can only be conducted on sequences of limited length, which may lead to discrepancies when applied to longer sequences. We observed such a

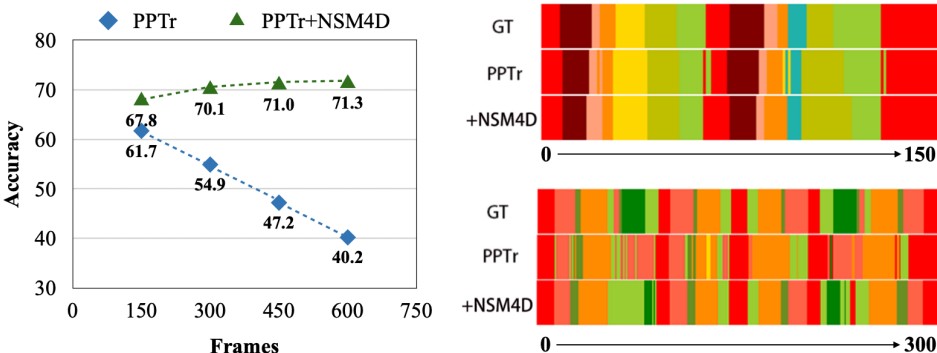

Figure 3: **Left**. As the inference sequence length increases, the performance of PPTr decreases significantly. However, incorporating NSM4D leads to further enhancements in performance. **Right**. Visualization results of PPTr and PPTr+NSM4D on action segmentation.

phenomenon in existing methods. However, our method exhibits exceptional scalability when it comes to handling long sequences, showcasing its superiority in this regard.

We train PPTr and PPTr+NSM4D using a sequence length of 150 and perform inference under varying sequence lengths. As depicted in the left of Figure 3, the performance of PPTr experiences a significant drop as the sequence length increases. However, incorporating NSM4D yields further enhancements in performance. This is primarily attributed to our core design, which involves factorizing the geometry and motion and dynamically updating the scene model. As a result, more information is effectively integrated to enhance perception in a structured manner. In contrast, existing methods tend to struggle when presented with noisy raw observations lacking proper alignment. The prediction results on HOI4D action segmentation depicted on the right side of Figure 3 offer a more intuitive illustration. PPTr+NSM4D demonstrates the ability to provide accurate and continuous results on long sequences, whereas PPTr alone often becomes confused and generates discontinuous results.

**Effectiveness of geometry and motion.** In our design, we employ geometry and motion factorization to capture distinct aspects of the dynamic scene. The geometry module is responsible for aggregating historical observations to construct a comprehensive scene representation. On the other hand, the motion module accumulates scene flow to comprehend object motion and camera ego-motion. When evaluating NSM4D with only the geometry module, the performance reaches 64.03, while using only the motion module yields a result of 65.91. Both of these scores are lower than the complete implementation's performance of 67.78. This observation highlights the crucial roles played by each component in the factorization process within our design.

Furthermore, the synergy between the geometry and motion components is evident, as a thorough understanding of geometry necessitates a robust comprehension of motion. Similarly, a more accurate understanding of motion relies on precise geometry correspondence. Therefore, the interplay between geometry and motion is essential for achieving a comprehensive understanding of the scene in an online manner.

## 6 CONCLUSION

In this paper, our focus lies in investigating the task of online 4D perception in general scenarios. We introduce NSM4D, a novel paradigm that can be seamlessly integrated into existing 4D backbones, thereby offering a substantial enhancement to their online perception capabilities. We apply our design to various backbones and models for both indoor and outdoor scenes. Extensive experiments and ablation studies demonstrate that NSM4D significantly boosts the online perception performance of the base models. We firmly believe that the generalizability of NSM4D enables its utilization in a diverse range of scenarios. The limitation is that we don't consider real-time applications so far, we aim to optimize NSM4D to achieve real-time inference speed in our future work.

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
