In this document, we provide a list of supplementary materials to support the main paper.

**More experiment and ablation results.** In Section A, we provide more experiment and ablation results to demonstrate our work's ability and superior property.

**Visualization.** In Section B, we provide some visualization results of NSM4D to visualize the effects of our approach.

**Implementation details.** We additionally provide implementation details in Section C.

**Limitations and future work.** In Section D, we discuss the limitations of our work. We also list possible directions for future work to explore.

## A  MORE EXPERIMENT AND ABLATION RESULTS

### A.1  SYNTHIA 4D DATASET

In this section, we provide a complete comparison with previous work on Synthia 4D(Ros et al., 2016) dataset. As shown in the table below, when introducing our method, constant improvement is shown over vanilla PPTr in an online manner, demonstrating that our neural scene model is indeed customized for online 4D perception. It further indicates that the compact and structured 4D history summary proposed in our paper is noticeably effective.

| Method | Frames | Bldn | Road | Sdwlk | Fence | Vegittn | Pole | Car | T.Sign | Pedstrn | Bicycl | Lane | T.Light | mIoU |
|--------|--------|------|------|-------|-------|---------|------|-----|--------|---------|--------|------|---------|------|
| PPTr | 3(offline) | **97.51** | 98.21 | **95.11** | **96.81** | **99.65** | 97.86 | **98.01** | 80.98 | **90.60** | 0.00 | 78.21 | 76.89 | 84.15 |
| PPTr | 3(online) | 94.68 | 98.32 | 93.11 | 95.26 | 97.30 | 98.06 | 95.42 | 82.74 | 85.55 | 0.00 | **78.31** | 81.70 | 83.37 |
| +NSM4D | 3(online) | 97.02 | **98.58** | 94.01 | 95.82 | 97.80 | **98.21** | 96.37 | **84.91** | 88.59 | 0.00 | 77.82 | **82.72** | **84.32** |

Table 1: Evaluation for online semantic segmentation on Synthia 4D dataset

### A.2  ROBUST TO NOISE.

Our neural scene model is updated with the guidance of scene flow estimated from point clouds. To enhance the robustness of our module to noise induced during the flow estimation process, we adopt the average flow of the points within a token when updating. This approach reduces the performance degradation caused by noise accumulation in the sequences. We conducted experiments to verify the robustness of our method, as flow estimation is a crucial aspect of our framework. Results show that our method is highly robust to noise induced from the estimated flow. Specifically, while using ground truth scene flow yields a performance 68.66, our method only has a 0.88% lower performance with the estimated flow.

### A.3  NUMBER OF TOKENS.

The number of tokens controls the granularity of dynamic modeling. Not enough tokens might only focus on some local regions, lacking the ability to represent the whole scene. Increasing the number of tokens can maintain more fine-grained details in the scene model, but it leads to more memory overhead and less robustness to random noise. We compare different numbers of tokens in the below table. Our method works best when the number of tokens is set to 64. Therefore, we choose 64 as the default number of our approach.

| Number | Frame-wise acc |
|--------|----------------|
| 32 | 66.14 |
| 64 | **67.78** |
| 128 | 66.53 |

## B  Visualization

In this section, we provide some visualization results to show the effectiveness of our approach. As

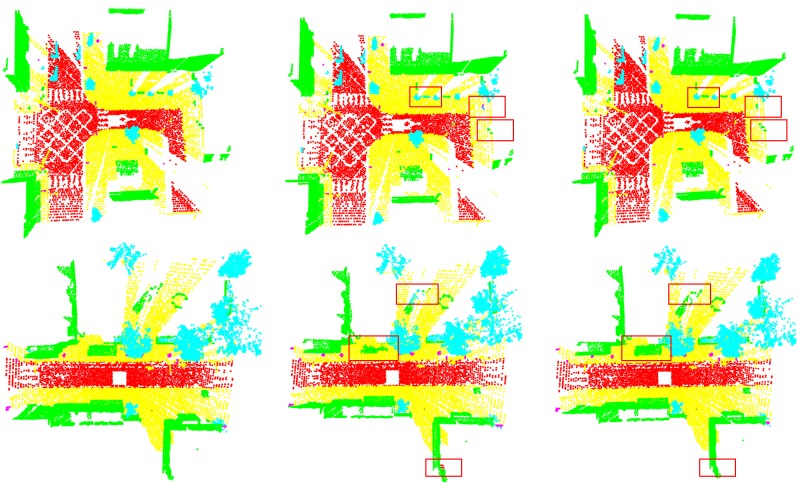

Figure 1: Visualization results on Synthia4D dataset.

shown in Figure 1, we use PPTr as the backbone model. From left to right is the ground truth label, segmentation results of vanilla PPTr and segmentation results of PPTr+NSM4D. The visualization results further shows that the proposed NSM4D is capable of aggregating geometry and motion information from the historical sequences.

## C  Implementation Details

This section introduces the details of implementing HOI4D action segmentation experiments.

We use SGD to train the neural network. The learning rate is set to be 0.02, and we use a warm-up learning rate scheduler for the first ten epochs, where the learning rate increases linearly. As for parameters, the feature dimension of the neural scene model is chosen to be 1024. For 4D convolution, we use ball query with a radius of 0.9, and the number of samples is set to 32. We adopt learning rate decay at 35, 60, and 80 epochs to achieve better performance by default. With batch size set to 8, our method can be implemented on two NVIDIA A100 40G when training with 150-frame sequences.

## D  Limitations and Future Work

Despite the impressive results, our work still has some limitations. First, NSM4D is customized for 4D online perception, but it is still unable to achieve real-time perception due to the computation overhead. We will reduce compute bottleneck and accelerate inference to get real-time dense perception. Also, the proposed neural scene model is purely point-based. If given RGBD sequences as input, how to leverage the fine-grained details in RGB sequences to form a more detailed geometry understanding and more accurate point correspondences in the neural scene model remains to be explored in the future.

## References

German Ros, Laura Sellart, Joanna Materzynska, David Vazquez, and Antonio M Lopez. The synthia dataset: A large collection of synthetic images for semantic segmentation of urban scenes. In *Proceedings of the IEEE conference on computer vision and pattern recognition*, pp. 3234–3243, 2016.