# OpenReview forum: "NSM4D: Neural Scene Model Based Online 4D Point Cloud Sequence Understanding"
_ICLR.cc/2024/Conference — ICLR 2024 Conference Withdrawn Submission_

### Official Review · Reviewer_C4KL · 2023-10-14

**Soundness:** 3 good
**Presentation:** 3 good
**Contribution:** 3 good
**Rating:** 6
**Confidence:** 1

**Summary:**

This paper proposes a new 4D point cloud sequence understanding method. The major techniques are 4D tokenization, token updating and token merging. The proposed approach demonstrated strong performance in HOI4D action segmentation and Semantic KITTI online semantic segmentation.

**Strengths:**

I'll update my scores after the rebuttal.

1. The proposed method demonstrated strong performance on two benchmarks. Moreover, their experimental results show that the proposed module can be used in various SOTA 4D backbones, which is very beneficial.

2. The proposed modules are somewhat novel and may inspire other works. I think the geometry and motion factorization is novel. I feel it is novel because it is not direct to see how this is implemented. I think the token operation modules might be insightful for future work.

3.  The writing is clear, and various analyses support this method.

**Weaknesses:**

1. I'm not familiar with this sort of work in action segmentation. Probably, some strong baselines are missing. Also, ICLR is probably not the best venue for this vision work. Also, the plug-and-play strategy may not be particularly novel in this area. I know PTv2 very well, but I am not sure whether this approach should compare to other methods for Semantic KITTI tasks, such as SphereFormer or other works. It seems that mIoU in Table 2 is pretty low (<55%).

2. Probably, this model is time-consuming because of the complex reasoning procedure.

3. Some parts of this submission are confusing to me:

(1) In the introduction, the authors say, "Querying a dynamic reconstruction for historical spatial-temporal contexts would be much more efficient compared with directly querying the raw 4D data." But I do not see this directly. From my experience, raw 4D data is fast to index and insert because of the regular 4D grid structures. More evidence is needed to support this fact. The authors may use some references, diagrams, or experiments to illustrate this. Also, this sentence connects poorly with other sentences.

(2) In the introduction, how the 4D sequence is factorized is unclear, but this is crucial in this method, I believe. Figure 1 is not that clear as well. The point cloud is messy, and the diagram is not informative.

(3) It is weird to say, "We propose a new **diagram** that serves as a **plug-and-play** framework" because a new diagram means a novel framework. Still, the plug-and-play module means a new module to put into existing frameworks. It's unsure whether this can be called a framework-level contribution.

(4) The problem statement part seems confusing. First, authors should provide mathematical symbols, which can lay the basis for their subsequent model description. Second, I'm unsure what "comparing online perception to offline perception" means. It seems that this is a minor technical detail. And why this is important is not explained. Furthermore, I cannot connect the challenges listed here to the introduction. It seems that the challenges introduced here are not the same as the introduction. Furthermore, too many problems are raised in this paper and the reader's attention is easily scattered.

**Questions:**

1. What is the number of parameters used in this model? How is the speed?

2. Can the proposed method generalize to other 3D scene representations?

---

### Official Review · Reviewer_KTef · 2023-10-28

**Soundness:** 3 good
**Presentation:** 3 good
**Contribution:** 3 good
**Rating:** 5
**Confidence:** 4

**Summary:**

This paper introduces NSM4D, a plug-and-play strategy designed to enhance existing 4D point cloud sequence perception backbones, particularly in their ability to understand long sequences with flexible length. The approach factorizes a 4D sequence into geometry and motion through implicit dynamic 3D reconstruction. The 4D tokenization module aggregates geometry and motion features from the current observation, with the token updating module using scene flow to align historical information with the current one. The token merging module manages new tokens and maintains a reasonable token size across different sequence lengths.

**Strengths:**

1. The paper is well-written and easily comprehensible.
2. The motivation behind the model design is reasonable, with tokenization offering a scalable approach for handling variable sequence lengths and efficient querying.
3. The plug-and-play design is effectively evaluated for certain tasks.

**Weaknesses:**

The major concern of the paper is the absence of a long-term task evaluation. This absence makes it challenging to ascertain the method's effectiveness in handling long-term sequence information, which is a crucial aspect of the paper's proposed approach. The evaluation primarily focuses on tasks that require short-term temporal information, leaving a significant gap in assessing the method's claims regarding its long-term sequence information retention and query capabilities.

Please see questions for more details.

**Questions:**

1. The depiction of the pipeline in Figure 1 is somewhat ambiguous and needs clarification for a more precise understanding of the processes involved.
2. While the method claims to handle long history sequences effectively, the evaluated tasks seem to focus on relatively short-term temporal information rather than requiring long-term information querying. It would be valuable to include a long-term task evaluation for a more comprehensive assessment.
3. An efficiency analysis is required to determine the additional time cost, FLOPs, and weights introduced by the proposed method.
4. The method relies on motion estimation from scene flow. Assessing the accuracy of this motion estimation is crucial. For instance, if ground truth motion is provided, how much improvement could be gained?
5. A discussion of the limitations is needed to identify the current design's bottlenecks and areas for improvement.
6. Additional visualizations of task performance evaluations would enhance the paper's comprehensibility.
7. In Figure 1 supplementation, the term "Synthia4D" is mentioned but not explained or introduced in the paper; further clarification is needed.

---

### Official Review · Reviewer_XSLL · 2023-10-30

**Soundness:** 2 fair
**Presentation:** 2 fair
**Contribution:** 2 fair
**Rating:** 5
**Confidence:** 2

**Summary:**

The paper introduces NSM4D, a new approach for understanding 4D point cloud sequences online.
NSM4D uses a neural scene model to factorize geometry and motion information, dynamically updating itself with new observations.
It improves over the baselines on online perception capabilities for both indoor and outdoor scenarios and shows robustness to sensor noise.

**Strengths:**

A straightforward component can be integrated into existing 4D frameworks.

Also, It shows an improvement over the baselines in terms of real-time perception for various scenarios and exhibits resilience to sensor disturbances.

**Weaknesses:**

1. The experiments results are a bit of confusion to me, and may be not sound or solid. The pptr segmentation results on HOI4D doesn't match the nubmers on pptr paper (pptr achieves 68.54 miou in the original paper with 3 frames, yet it only achieves 42.73 in Table3). Similarly, P4Transformer is also way off. There isn’t clear explanation provided for these discrepancies (except the flip-stitch, which unlikely to cause this big difference). To evaluate the effectiveness of the proposed methods, it’s crucial to compare performance with the baselines from the original papers. If there are changes in the settings or evaluation methods, these should be clearly stated and justified. Please correct me if I’m wrong, as I have not previously worked on HOI4D.
4. The speed of the method is a concern. Given that RAFT3D need to applied to obtain the flow of past frames, I anticipate that the process would be quite slow. This is a critical issue for a method termed “online” scene understanding, which typically necessitates high efficiency.
2. The novelty of the work is somewhat questionable. The concept of “decoupling geometry and motion” doesn’t resonate with me as it seems to be just a feature representation with explicit location.
3. The writing is a bit wordy and not that clear. e.g. the beginning of Section 4 (NEURAL SCENE MODEL) is almost just a repeat of the introduction.

**Questions:**

Please provide clarification on the experiment numbers mentioned in the weakness section.

 It would be greatly appreciated if you could use the settings of the baselines (e.g., the settings mentioned in table 2 of the pptr paper) for a clearer comparison. Additionally, it would be helpful to have a more precise comparison with ptv2 on semantickitti. Ideally, please consider submitting your results to the test server.

---

### Official Review · Reviewer_Usdf · 2023-11-01

**Soundness:** 1 poor
**Presentation:** 1 poor
**Contribution:** 1 poor
**Rating:** 3
**Confidence:** 4

**Summary:**

This paper proposes an online 4D feature extractor that takes a 3D pointcloud sequence as an input and output set of tokens for each frame in an online fashion. The model needs to take scene flow predictions as input and then explicitly track points and aggregate features. The author claimed that by combining their model with current 3D perception backbones, they can achieve state-of-the-art segmentation results on HOI4D and SemanticKITTI.

**Strengths:**

The versatility of its design, making it adaptable to most state-of-the-art 3D backbones, is one strength.

**Weaknesses:**

- The presentation of this paper needs to be improved.
    - This paper contains lots of cluttered sentences. Lots of the sentences in this paper are long and sometimes meaningless, making readers hard to follow. It would be beneficial to rewrite lots of it more concisely. For example, the first paragraph in section 4: "Our goal is to devise a methodology that is both effective and efficient for sequentially aggregating long-term historical information for precise 4D online perception. NSM4D aims to achieve this by introducing a generic paradigm design that enables seamless integration with the existing 4D networks, empowering them with the capability of efficient and effective online 4D perception."    This paragraph can be rewritten in a clearer and more concise way: We devise a method to aggregate 3D pointcloud sequences online called NSM4D. NSM4D can integrate with existing 4D networks seamlessly for online 4D perception.
    - Figures are hard to understand.
        - Figure 1 is referenced in Section-1, but it is hard to understand without finishing reading Section-4. Moreover, there isn't too much useful information in Figure-1.
        - Figure-2 has lots of information, but it is still hard to understand every component. Also, it would be beneficial to add (a), (b), (c) under three parts of Figure-2.
- Experimental evaluation is not solid. There is a problem in Table 2. I checked the SemanticKiTTI leaderboard, https://paperswithcode.com/sota/3d-semantic-segmentation-on-semantickitti, and Table 2, 3 of paper SPVCNN. The number of baseline methods need to be corrected. For example, the mIoU of SPVCNN is above 60% in their table-2,3. And PTv2 achieved around 70% mIoU according to the leaderboard. However, this paper only reports mIoU of 49.70% and 51.13% for SPVCNN and PTv2, respectively. Also, experiments on more datasets would be beneficial.

**Questions:**

- There are some missing details about the method.
    - How many points are sampled at every frame.
    - Why use uniform sampling, rather than furthest points sampling, or stratified sampling?
    - Size and computational cost of the model.
- This paper uses the word: "efficient" to describe their methods several times. It would be beneficial to provide some analysis and comparisons on running speed or computational cost to justify this.

---

### Comment · Reviewer_C4KL · 2023-11-18
**No rebuttal?**

Is this true that this paper will not post a rebuttal?

---

### Author Response · Authors · 2023-11-18
**General Response**

We appreciate all the reviewers for their insightful and constructive feedback. We would like to provide a general response here that encapsulates some of the major points addressed.

- Experimental evaluation for semantic KITTI

The cited results in Table 2 come from [MarS3D](https://openaccess.thecvf.com/content/CVPR2023/papers/Liu_MarS3D_A_Plug-and-Play_Motion-Aware_Model_for_Semantic_Segmentation_on_Multi-Scan_CVPR_2023_paper.pdf) and we simply follow the results on validation set for different models shown in their table. Our experimental setting remains the same as described in their paper: we select the multi-scan task as the validation benchmark, which includes 25 classes distinguishing static and moving objects. We report mIoU for each class on validation set.  Also, for PTv2, it simply has results on single-scan task, so we do some modifications and implement a multi-scan version of PTv2.

- Experimental evaluation for PPTr

We checked the results of PPTr on HOI4D dataset with its authors and found that they only use part of the dataset and the labels are outmoded. So to correctly evaluate our model, we follow the setting described in [C2P](https://openaccess.thecvf.com/content/CVPR2023/html/Zhang_Complete-to-Partial_4D_Distillation_for_Self-Supervised_Point_Cloud_Sequence_Representation_Learning_CVPR_2023_paper.html) and report results shown in their table.

- Long-term task evaluation

We provide long-term task evaluation for HOI4D Action Segmentation task. We use sequences with a length of 600 frames to demonstrate the ability of our method. As shown in the analysis section, our method exhibits exceptional scalability when it comes to handling long sequences.

- Efficiency for online scene understanding

We will add more experiments to show the efficiency of our method. However, the proposed method is not aimed at real-time 4D scene understanding, and there are faster and more accurate flow estimators. As our method is robust to noise introduced by flow estimation and compatible with different kinds of flow estimators, we can simply adopt our method with faster flow estimators and further improve the overall efficiency.